# Maternal serum progesterone and BMI are associated with neonatal sex ratios following single frozen embryo transfer

Robert Czech[1], Dariusz Wójcik[1], Tomasz Skweres[1], Wojciech Śliwiński[1], Dorota Zamkowska[1], Przemysław Ciepiela [2]*

**1** Gameta Gdynia Center of Health, Gdynia, Poland, **2** Bocian Fertility Clinic, Gynecology and Obstetrics, Szczecin, Poland

* przemyslaw.ciepiela@gmail.com

## Abstract

### Background

The biological mechanisms behind sex ratio variation in assisted reproductive technologies (ART) are not well understood. This study investigated whether maternal serum progesterone (PRG) and body mass index (BMI) are related to neonatal sex after frozen–thawed embryo transfer (FET).

### Methods

We conducted a retrospective cohort study of 998 single blastocyst FET cycles performed under a freeze-all strategy from 2019 to 2023. Maternal BMI and serum PRG measured in the peri-transfer period were analyzed in relation to neonatal sex. Receiver operating characteristic (ROC) analyses identified predictive thresholds. Associations were evaluated using multivariable logistic regression models adjusted for potential confounders.

### Results

ROC analyses identified PRG ≤ 21.11 ng/mL and BMI ≤ 21.30 kg/m² as the optimal cut-offs for predicting male birth. Low PRG was associated with a higher odds of male offspring (OR = 1.61; 95% CI: 1.09–2.36; p = 0.016), as was low BMI (OR = 2.33; 95% CI: 1.61–3.40; p < 0.001). When both factors were present, the likelihood of male birth increased further (OR = 2.30; 95% CI: 1.39–3.80; p = 0.001). A high BMI reduced the association between low PRG and male birth, while an elevated PRG weakened the connection between low BMI and neonatal sex. Female births were correspondingly less common under these conditions (OR = 0.43; 95% CI: 0.26–0.72; p = 0.001).

**Data availability statement:** The minimal anonymized dataset underlying the results of this study is available in the Zenodo repository: https://doi.org/10.5281/zenodo.19314941 The dataset includes all variables necessary to reproduce the analyses presented in this study. All data have been fully anonymized in accordance with applicable data protection regulations. No additional data are available.

**Funding:** The author(s) received no specific funding for this work.

**Competing interests:** The authors have declared that no competing interests exist.

## Conclusions

Maternal progesterone levels and BMI together were associated with neonatal sex ratios after FET. These findings imply that endocrine and metabolic environments influence embryo–endometrium interactions in a sex-specific way and open new pathways for research into developmental programming in ART. These findings do not establish causality and should be interpreted as hypothesis-generating.

## Introduction

Assisted reproductive technologies (ART) have made significant progress in recent decades, greatly enhancing outcomes for infertility treatments and broadening our understanding of early human development. Innovations in embryology labs, such as time-lapse embryo monitoring, have transformed in vitro fertilization (IVF) from just a clinical procedure into a valuable tool for studying human reproductive biology [1,2]. Variation in neonatal sex ratios has become an unexpected aspect of assisted reproductive technologies (ART), raising important questions about embryo development and maternal biology [3,4]. Although live birth rates remain the main indicator of ART success, subtle differences in the secondary sex ratio (SSR), the ratio of male to female live births, have been observed [3–6]. These changes are not only biologically interesting but also clinically significant, as neonatal sex has been associated with perinatal outcomes and long-term health risks [3,5–7].

Several ART-related factors seem to influence SSR. Blastocyst transfer, especially of high-quality embryos, has been linked to a higher rate of male births, while intracytoplasmic sperm injection (ICSI) may reduce this trend or even promote female births, particularly when lower-grade trophectoderm or expansion scores are involved [3,4,6,8]. These observations suggest that embryo handling and developmental dynamics may have sex-specific effects, possibly through epigenetic mechanisms with long-lasting consequences.

Embryonic contributions alone, however, cannot fully account for the observed variation in sex ratios. Male embryos are sometimes reported to reach the blastocyst stage more efficiently but may carry more chromosomal abnormalities [6], while female embryos may show higher rates of aneuploidy [5,9]. Importantly, implantation and ongoing pregnancy rates appear to be largely unaffected by embryonic sex. Since the primary sex ratio (PSR) at fertilization is approximately equal, deviations in SSR are likely influenced during post-fertilization development, implantation, or very early pregnancy [5]. These findings highlight the potential role of maternal physiology in affecting neonatal sex. Because the present analysis was restricted to singleton live births, implantation failure and very early pregnancy loss were not directly observable; therefore, secondary sex ratio at birth was used as a proxy for sex-specific survival during implantation and early gestation.

Among maternal characteristics, body mass index (BMI) has emerged as a potential factor influencing offspring sex. Our previous research showed that higher

maternal BMI was associated with a greater likelihood of female births [10]. This suggests that metabolic factors might impact the intrauterine environment in ways that affect sex-specific survival or implantation. At the same time, progesterone (PRG) plays a vital role in implantation and endometrial receptivity, and experimental data suggest that sex-specific responses to maternal hormones could influence embryo viability [7,11–14]. However, the combined effects of BMI and progesterone on SSR in ART have not been extensively studied.

In this retrospective cohort study, we examined the association between maternal BMI, serum progesterone levels, and neonatal sex following frozen–thawed embryo transfer. To minimize confounding factors, only the first FET cycles were included, using a fully artificial, hormone-controlled endometrial preparation to eliminate the impact of ovarian stimulation. Cycles involving preimplantation genetic testing (PGT) were excluded to preserve natural embryo selection processes. By isolating these variables, our goal was to provide new insights into how metabolic and hormonal maternal environments influence embryo – endometrium interactions and, ultimately, neonatal sex outcomes in ART.

## Materials and methods

### Study design and participants

This retrospective cohort study analyzed 408 singleton live births from 998 consecutive frozen-thawed single embryo transfer cycles conducted between 2019 and 2023 at a tertiary fertility center. Because genetic sex is established at fertilization and the primary sex ratio is approximately equal, maternal BMI and progesterone levels at fertilization are not expected to influence sex determination; therefore, exposures were defined at the time of embryo transfer to examine factors potentially associated with sex-specific implantation and early survival contributing to secondary sex ratio variation. All transfers employed a freeze-all strategy, utilizing top-quality embryos selected through standardized morphological grading. Only the first frozen embryo transfer per patient was included to avoid within-patient correlation, cumulative exposure bias, and embryo selection effects associated with multiple transfers. Exclusion criteria included: oocyte donation, and maternal risk factors for recurrent pregnancy loss, such as genetic abnormalities, antiphospholipid syndrome, uterine synechiae, or Müllerian malformations. Cycles involving preimplantation genetic testing (PGT) were excluded to preserve natural embryo selection and implantation dynamics, as PGT-based designs address complementary but methodologically distinct research questions. Clinical and embryological data were collected from the electronic medical record system (i-IVC database) on February 15, 2025. Before analysis, the dataset was fully anonymized to ensure that investigators could not access any information that would directly identify individual participants, both during and after data collection. Ethical approval was obtained from the Bioethics Committee of the District Medical Chamber in Gdańsk, Poland (Ref. No. KB-43/21).

### Ovarian stimulation, embryo culture, and vitrification

Controlled ovarian stimulation (COS) used either a GnRH agonist (Gonapeptyl 0.1 mg; Ferring Pharmaceuticals, Germany) or antagonist (Cetrotide 0.25 mg; Merck Serono, Germany) protocol, combined with recombinant FSH (150–225 IU; Gonal-F, Merck Serono, Switzerland; or Puregon, Organon, Netherlands), titrated to baseline anti-Müllerian hormone (AMH) levels. Final oocyte maturation was triggered with recombinant hCG (250 µg; Ovitrelle, Merck Serono, France) or GnRH agonist (0.2 mg Gonapeptyl). All oocytes underwent intracytoplasmic sperm injection.

Embryos were cultured to the blastocyst stage and graded per the Istanbul consensus [15], assessing expansion (Grades 1–6), inner cell mass (ICM: A–C), and trophectoderm (TE: A–C). All morphological assessments were performed and documented by experienced senior embryologists in accordance with routine laboratory quality standards. Blastocysts were vitrified using Kitazato VT601 media and warmed with VT602 media per the manufacturer's protocol, 2–3 h before transfer. Single embryo transfer was performed regardless of maternal age or reproductive history.

## Endometrial preparation and luteal support

All FET cycles used a standardized hormone replacement therapy protocol. Estradiol (2 mg, Estrofem; Novo Nordisk Healthcare AG, Switzerland) was administered intravaginally twice daily from cycle day 2. The vaginal route was selected as part of the center's standardized protocol to ensure consistent endometrial exposure while minimizing first-pass hepatic metabolism. Transvaginal ultrasound on cycle days 13–15 confirmed endometrial thickness.

Patients with a trilaminar endometrium ≥8 mm initiated luteal support with micronized vaginal progesterone (600 mg daily; Luteina, Adamed, Poland) plus oral dydrogesterone (30 mg daily; Duphaston, Solvay Pharmaceuticals, UK). It continued through embryo transfer, scheduled on day 5 of progesterone administration (14:00–15:00). Serum progesterone was measured 1–2 h before transfer, maintaining a consistent interval from the morning dose (6:00–7:00). Luteal support was continued until serum β-hCG testing at 12–14 days, and in confirmed pregnancies, up to 10–12 weeks of gestation. Intrauterine pregnancy was confirmed by ultrasound at 8 weeks.

## Study variables and outcomes

The primary outcome was neonatal sex (male vs. female) at delivery. The main exposures were maternal serum progesterone concentration (ng/mL) on the day of embryo transfer and maternal BMI (kg/m²). Secondary variables included maternal age, AMH concentration, sperm concentration, and progressive motility (WHO 5th edition, Grade A), endometrial thickness, and embryo quality (Gardner grading). Male factor parameters were included as covariates in adjusted analyses; however, existing evidence indicates that male factor infertility has a limited influence on secondary sex ratio at live birth, particularly in ICSI cycles where fertilization mechanics are standardized [9].

## Statistical analysis

Categorical variables were summarized as counts and percentages; continuous variables as medians and interquartile ranges (IQR). Missing data were excluded. Receiver operating characteristic (ROC) analysis identified optimal cut-off values for progesterone and BMI, using male live birth as the positive outcome [Hanley & McNeil, 1982]. Logistic regression models adjusted for confounders estimated associations. Advanced methods were applied to strengthen causal inference:

- Targeted Maximum Likelihood Estimation (TMLE) to assess exposure effects,

- Inverse Probability Weighting (IPW) to reduce selection bias,

- E-value analysis to test sensitivity to unmeasured confounding, and

- Placebo testing to detect residual bias.

Based on ROC-derived thresholds, five exposure models were defined: Model 1 (progesterone ≤ 21.11 ng/mL, any BMI), Model 2 (BMI ≤ 21.30 kg/m², any progesterone), Model 3 (progesterone ≤ 21.11 ng/mL and BMI ≤ 21.30 kg/m²), Model 4 (progesterone ≤ 21.11 ng/mL and BMI > 21.30 kg/m²), and Model 5 (progesterone > 21.11 ng/mL and BMI ≤ 21.30 kg/m²). These groupings were used for exploratory causal modeling rather than for clinical classification. All analyses were performed in R Statistical Software (v4.3.3; R Core Team, 2024) under Windows 11 Pro (build 26100).

## Results

### Baseline characteristics

A total of 408 singleton neonates were analyzed, originating from 998 consecutive frozen-thawed single embryo transfer cycles conducted between 2019 and 2023. Maternal age at FET ranged from 23 to 44 years (median 33.0, IQR 30.0–36.0), while maternal BMI varied from 16.9 to 43.9 kg/m² (median 22.4, IQR 20.5–25.2). Anti-Müllerian hormone levels ranged from 0.24 to 30.0 ng/mL (median 3.63, IQR 2.40–5.57). Semen parameters were variable, with a median sperm concentration of

25.8 million/mL (IQR 7.2–59.6) and progressive motility (Grade A) at 19% (IQR 9.0–30.0). At the start of progesterone treatment, endometrial thickness measured between 6 and 14 mm (median 10, IQR 9–11). Embryo quality was consistently high, with 93.9% expansion Grade 4, 82.8% ICM Grade A, and 80.6% TE Grade A. The neonatal sex ratio was balanced, with 197 males (48.3%) and 211 females (51.7%), providing sufficient power for sex-stratified analyses (Table 1).

## Thresholds for progesterone and BMI

Receiver operating characteristic analyses identified serum progesterone ≤ 21.11 ng/mL (AUC = 0.56) and BMI ≤ 21.30 kg/m² (AUC = 0.58) as optimal thresholds for predicting male birth. These AUC values reflect limited discriminative ability, and the thresholds are not used for clinical diagnosis. Instead, they serve to define biologically plausible exposure groups for causal modeling.

## Independent associations of progesterone and BMI

In Model 1, low serum progesterone (≤ 21.11 ng/mL) was significantly linked to male birth (RR = 1.28, 95% CI: 1.04–1.58, p = 0.018; OR = 1.61, 95% CI: 1.09–2.36, p = 0.016) (Table 2, Fig 1). Average treatment effect analyses supported this

**Table 1. Baseline characteristics of the study population (n = 408).**

| Characteristic | n | Value |
|---|---|---|
| PRG 5 days post-FET (ng/mL) | 408 | 20.40 (15.65–25.38)[1] |
| Neonate sex: | 408 | |
| - Boy | | 197 (48.28%)[2] |
| - Girl | | 211 (51.72%)[2] |
| Maternal age at FET (years) | 408 | 33.00 (30.00–36.00)[1] |
| Maternal BMI (kg/m²) | 408 | 22.40 (20.50–25.23)[1] |
| Maternal AMH (ng/mL) | 408 | 3.63 (2.40–5.57)[1] |
| Semen Concentration (million/mL) | 408 | 25.80 (7.20–59.60)[1] |
| Semen Motility Grade A (%) | 408 | 19.00 (9.00–30.00)[1] |
| Endometrial Thickness (mm) | 408 | 10.00 (9.00–11.00)[1] |
| Embryo 1 expansion grade: | 408 | |
| - Grade 1 | | 1 (0.25%)[2] |
| - Grade 2 | | 1 (0.25%)[2] |
| - Grade 3 | | 21 (5.15%)[2] |
| - Grade 4 | | 383 (93.87%)[2] |
| - Grade 5 | | 2 (0.49%)[2] |
| Embryo 1 inner cell mass (Node): | 408 | |
| - Grade 1 | | 338 (82.84%)[2] |
| - Grade 2 | | 70 (17.16%)[2] |
| Embryo 1 trophoectoderm grade: | 408 | |
| - Grade 1 | | 329 (80.64%)[2] |
| - Grade 2 | | 75 (18.38%)[2] |
| - Grade 3 | | 4 (0.98%)[2] |

(Values presented as median [IQR] or number [%]).

Note: [1] data are presented as median (IQR), reflecting the 25th and 75th percentiles; [2] data are presented as n (%), where n is the number of observations in the category and % is the percentage of the total sample. Abbreviations: PRG = Progesterone; FET = Frozen Embryo Transfer; BMI = Body Mass Index; AMH = Anti-Müllerian Hormone. Embryo Grading: Expansion Grade, Inner Cell Mass, and Trophectoderm Grade are based on standard embryological scoring systems.

**Table 2. Stratified models assessing the likelihood of male neonatal birth after frozen embryo transfer (FET).**

| Model | Definition | Male newborns (n) | Female newborns (n) | Relative risk (95% CI) | p-value | Odds ratio (95% CI) | p-value |
|---|---|---|---|---|---|---|---|
| 1 | PRG ≤ 21.11 (any BMI) | 122 | 103 | 1.28 (1.04–1.58) | 0.018 | 1.61 (1.09–2.36) | 0.016 |
| 2 | BMI ≤ 21.30 (any PRG) | 90 | 61 | 1.51 (1.26–1.81) | <0.001 | 2.33 (1.61–3.40) | <0.001 |
| 3 | PRG ≤ 21.11 and BMI ≤ 21.30 | 51 | 27 | 1.46 (1.19–1.78) | <0.001 | 2.30 (1.39–3.80) | 0.001 |
| 4 | PRG ≤ 21.11 and BMI > 21.30 | 71 | 76 | 0.94 (0.77–1.17) | 0.621 | 0.90 (0.60–1.36) | 0.619 |
| 5 | PRG > 21.11 and BMI ≤ 21.30 | 39 | 34 | 1.23 (0.97–1.57) | 0.088 | 1.54 (0.90–2.61) | 0.113 |

**Abbreviations:** PRG, serum progesterone level on the day of embryo transfer; BMI, body mass index; CI, confidence interval. Relative risks and odds ratios refer to the likelihood of male neonatal sex.

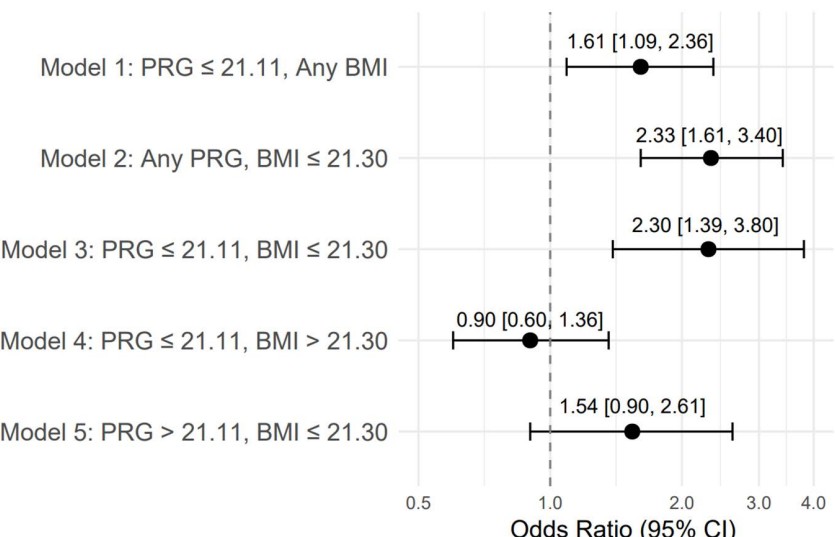

**Fig 1. Odds ratios (OR) for male birth according to maternal serum progesterone and BMI strata (Models 1–5).** Error bars represent 95% confidence intervals.

result (ATT = 0.13, p = 0.009; ATC = 0.10, p = 0.041), and bootstrap confidence intervals confirmed the robustness of the findings. In Model 2, low maternal BMI (≤ 21.30 kg/m²) independently increased the likelihood of male birth (RR = 1.51, p < 0.001; OR = 2.33, 95% CI: 1.61–3.40, p < 0.001). The effect was further supported by strong average treatment estimates (ATT and ATC = 0.21, p < 0.001).

## Concurrent exposure to low progesterone and low BMI

In Model 3, the simultaneous presence of low PRG and low BMI was linked to the highest likelihood of male birth (Table 2, Fig 1). The expected probability of a male neonate was 65% with exposure compared to 45% without exposure (ATE = 0.20, p = 0.001), resulting in an OR of 2.30 (95% CI: 1.39–3.80, p = 0.001). These findings were consistent across bootstrap and influence-curve analyses.

## Progesterone – BMI interplay in discordant conditions

Model 4 assessed low PRG in women with high BMI. No significant link to neonatal sex was found (RR = 0.94, p = 0.621; OR = 0.90, p = 0.619), indicating that a higher BMI may reduce the impact of low progesterone levels. Model 5 showed high PRG in women with low BMI. There was a non-significant trend toward a higher chance of male births (OR = 1.54, p = 0.113), but the result was not statistically significant.

## Sex-specific inversion

Analyses from the female offspring perspective demonstrated a complementary pattern. The conditions favoring male births (low PRG and low BMI, Model 3) were associated with a decreased probability of female births (OR = 0.43, 95% CI: 0.26–0.72, p = 0.001) (Table 2, Fig 2). This sex-specific inversion underscores the biological plausibility of the findings. Absolute numbers of male and female neonates within each exposure stratum are presented in Table 2.

Visual representations of the model-based odds ratios for male and female births are shown in Figs 1 and 2, respectively.

## Discussion

This retrospective cohort study investigated the relationships between maternal serum progesterone levels, body mass index, and neonatal sex following frozen – thawed embryo transfer. By isolating these variables within a standardized frozen-embryo transfer setting, our aim was to generate hypotheses about how maternal metabolic and hormonal environments may relate to embryo–endometrium interactions and subsequent variation in the secondary sex ratio, rather than to imply the determination of genetic sex. Both factors were associated with neonatal sex outcomes, and their joint presence was associated with higher odds of male births. Low PRG and low BMI were each independently associated with higher odds of male offspring (Models 1 and 2); when both were present, the odds were highest (Model 3). Conversely, high BMI appeared to lessen the impact of low PRG (Model 4), while high PRG in lean women showed only a non-significant trend

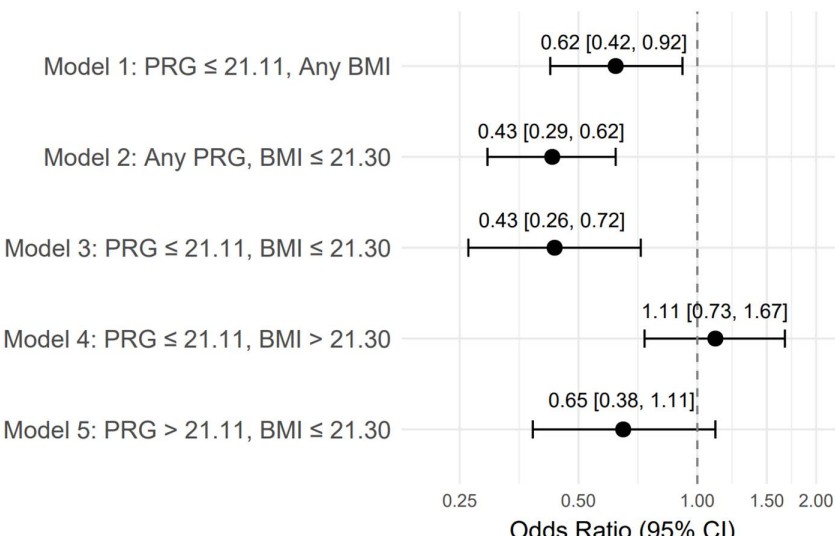

**Fig 2. Odds ratios (OR) for female birth according to maternal serum progesterone and BMI strata (Models 1–5).** Error bars represent 95% confidence intervals.

toward male births (Model 5). Overall, these findings suggest that maternal hormonal and metabolic environments may influence implantation outcomes in a sex-specific manner.

### Progesterone and Endometrial Receptivity

Progesterone is the primary regulator of endometrial transformation from a proliferative to a receptive state, enabling the establishment of the window of implantation (WOI). In artificial frozen embryo transfer cycles, where corpus luteum – derived hormones are absent, exogenous progesterone alone controls endometrial maturation. Several studies have shown that low serum progesterone levels on the day of transfer are linked to lower implantation and live birth rates [16–18]. Conversely, excessive or untimely progesterone exposure can lead to a premature WOI, disrupting synchrony between the endometrium and the embryo by affecting gene expression, cytokine signaling, and the availability of adhesion molecules [19]. Emerging data also indicate that progesterone can cause epigenetic modifications in endometrial cells, potentially affecting receptivity at the molecular level [19]. Accordingly, progesterone concentrations measured at the time of embryo transfer are biologically relevant to implantation dynamics and early embryonic survival, whereas progesterone levels at fertilization would not be expected to influence genetic sex determination or subsequent secondary sex ratio outcomes.

In our study, higher serum progesterone levels were independently linked to a greater likelihood of female births (aOR = 1.61; 95% CI, 1.09–2.36). This adds a new dimension to the known role of progesterone in implantation by suggesting that the hormonal environment may affect reproductive outcomes differently based on sex. The finding aligns with existing evidence of inherent developmental and metabolic differences between male and female embryos. Previous research has documented sex-related differences in trophoblast invasion, growth rates, timing of implantation, and response to maternal hormonal signals [20–22]. These differences suggest that embryos of different sexes may respond differently to maternal progesterone levels at the time of implantation. Although direct mechanistic data are limited, our results raise the possibility that circulating progesterone concentrations within physiologically relevant ranges for FET give rise to speculation that circulating progesterone concentrations within physiologically relevant ranges for FET may differentially relate to implantation or early survival of male and female embryos. Further research combining endocrine profiling with molecular analysis of the implantation site is needed to determine whether these early embryo–endometrium interactions are affected by fetal sex.

### Maternal BMI and Neonatal Sex

Maternal BMI also emerged as an independent factor influencing neonatal sex. Women with a BMI ≤ 21.30 kg/m² had a significantly higher chance of delivering male offspring (aOR = 2.33; 95% CI, 1.61–3.40), whereas higher BMI was associated with an increased likelihood of female births. These findings build on our previous observations in a large cohort of single embryo transfers (n = 2679), where we reported an inverse relationship between BMI and the probability of male births in frozen – thawed cycles [10]. Although the dataset partially overlaps with a previously published cohort, the present study is an independent analysis of an expanded population limited to first frozen embryo transfer cycles and includes maternal serum progesterone concentration as a novel exposure. This design enhances cohort homogeneity and reduces cumulative transfer and embryo selection bias. The current results confirm and refine that association, indicating that maternal metabolic status may have sex-specific effects on embryo implantation or early survival.

Although relatively few studies have examined the relationship between BMI and offspring sex, extensive evidence suggests that BMI is linked to reproductive outcomes in assisted reproduction. Elevated BMI has consistently been associated with reduced implantation and live birth rates [23,24], likely through impaired endometrial receptivity, chronic low-grade inflammation, and altered endocrine signaling. Tarín et al. proposed that the maternal metabolic environment could influence embryo selection or survival in a sex-dependent manner [25], an interpretation consistent with our results. The scarcity of comparable data from natural conception cohorts may reflect the confounding effect of spontaneous early losses, which are less observable in ART populations where implantation events are directly monitored.

These findings collectively reinforce that maternal BMI not only influences overall ART success but may also affect the neonatal sex ratio. The exact mechanisms are still unknown and may involve pathways related to energy balance, inflammatory mediators, and steroid metabolism, which differentially influence male and female embryonic development.

## Progesterone – BMI Interaction and Implantation

The combined impact of progesterone and BMI proved more informative than either factor alone. The strongest association with male births was observed when low serum progesterone levels coincided with a low maternal BMI, with an adjusted odds ratio of 2.30 (95% CI, 1.39–3.80). This suggests that the simultaneous presence of low progesterone and low BMI may indicate a unique endocrine – metabolic profile that affects the peri-implantation environment in a sex-dependent manner. Although causality cannot be determined, evidence from basic and clinical research supports the biological plausibility of an interaction between progesterone signaling and adiposity.

Experimental studies suggest that progesterone can influence leptin and appetite-related pathways, increasing food intake and fat accumulation [26]. Conversely, a higher BMI may change progesterone pharmacokinetics and tissue exposure by increasing distribution volume, altering liver metabolism, or sequestering in fat tissue [27,28]. Clinical observations report lower circulating progesterone levels in women with obesity undergoing IVF, which could impair optimal endometrial development [28,29]. Beyond blood levels, Mauland et al. found increased progesterone receptor expression in endometrial cancer tissues of women with higher BMI, indicating that adiposity may also affect how the endometrium responds to progesterone [30]. Similar hormonal changes have been seen in men, where obesity is linked to reduced progesterone and disrupted steroid balance [31]. Conversely, Shen et al. demonstrated a positive correlation between BMI and progesterone during pregnancy, showing that this relationship depends on the physiological state and timing of measurement [32]. The study by Bellver et al. found that obese women had significantly lower progesterone concentrations on embryo transfer day compared to non- obese women, possibly linked to hormonal imbalances, including secondary decreases in luteinizing hormone, which may impair progesterone production [29]. Studies by Lim et al., Whynott et al., and Demirel et al. support the association between BMI, progesterone levels, and reproductive outcomes, emphasizing the importance of considering maternal BMI and hormonal status in clinical protocols [27,28,33].

Together, these studies illustrate a complex, bidirectional relationship between progesterone and adiposity that may extend to the biology of implantation. In our cohort, a higher BMI appeared to weaken the association between progesterone and neonatal sex. In contrast, lean women with high progesterone levels showed only a nonsignificant trend toward male births. These results suggest that maternal BMI might influence the functional effect of progesterone on implantation success, potentially in a sex-specific way. However, they should be viewed cautiously and considered as hypothesis-generating rather than conclusive. Although implantation rates of embryos with known sex (e.g., following PGT-A) could provide direct evidence of sex-specific implantation differences, such designs involve embryo biopsy and selection, which may substantially alter implantation dynamics and mask spontaneous selection processes. The present study therefore focused on non-PGT frozen embryo transfer cycles to preserve unmanipulated embryo–endometrium interactions. In this context, the neonatal sex ratio at live birth serves as a pragmatic proxy outcome reflecting the cumulative result of sex-specific implantation success and very early pregnancy survival. Accordingly, our findings should be viewed as hypothesis-generating and complementary to future PGT–A–based studies designed to directly test sex-specific implantation probabilities. Finally, the discriminative power of progesterone and BMI for predicting neonatal sex was limited (AUCs 0.56 and 0.58, respectively). This limitation suggests poor diagnostic performance but does not rule out biological significance in multifactorial processes, such as implantation. In our study, ROC-based stratification was used as an exploratory tool to identify biologically plausible subgroups for targeted causal inference, rather than to establish clinical cut-offs.

## Strengths and limitations

This study is among the first to examine the combined relationship among maternal serum progesterone, body mass index, and neonatal sex following frozen single-embryo transfer. Strengths include a uniform population limited to first FET cycles with single blastocyst transfers and standardized endometrial preparation; sex-blinded embryo selection based solely on morphology; and the use of advanced statistical and causal inference methods, Targeted Maximum Likelihood Estimation, Inverse Probability Weighting, E-values, and placebo testing, to improve robustness and reduce confounding.

Limitations include the retrospective observational design, which precludes causal inference; the restriction to programmed cycles, thereby excluding natural and stimulated cycles with different hormonal dynamics; and a single serum progesterone measurement on the transfer day, which limits understanding of temporal fluctuations and overall exposure. Although male factor parameters were adjusted for, residual confounding related to sperm characteristics cannot be entirely excluded; however, their contribution to variation in the secondary sex ratio is considered limited in the context of single-blastocyst ICSI cycles. Dydrogesterone, a synthetic progestin used for luteal support, could not be quantified because it is not detectable by standard serum progesterone assays. Although it contributes to overall progestational support, it does not affect serum progesterone levels and was therefore excluded from the exposure definition. Serum progesterone is an imperfect proxy for endometrial hormonal status but remains the most practical and clinically accessible marker of luteal function. Although dydrogesterone and its active metabolite (DHD) do not cross-react with serum progesterone assays, they exert biologically relevant progestational effects at the endometrial level. Emerging evidence suggests that BMI-related metabolic variability may influence dydrogesterone/DHD exposure, raising the possibility that unmeasured variation in progestational activity could partially confound or modify the observed associations among maternal BMI, measured serum progesterone, and neonatal sex ratio. This potential influence could not be directly quantified in the present study and should be considered when interpreting the findings [34]. Finally, ROC-derived AUCs for progesterone and BMI were modest (0.56 and 0.58), indicating limited predictive ability; however, our analyses aimed to examine biologically informed associations rather than clinical prediction, and we identified statistically significant effects within our causal inference framework. Importantly, this study neither endorses nor encourages any form of sex selection or intentional modification of maternal physiology to determine fetal sex. It aims to enhance understanding of the natural interactions between maternal endocrine and metabolic factors and embryonic development in assisted reproduction.

## Conclusions

In this retrospective analysis of frozen embryo transfer cycles, we found significant links between maternal serum progesterone levels, body mass index, and neonatal sex. Male births were more frequently observed among pregnancies with low progesterone and low BMI, while female births were more frequently observed among pregnancies with higher progesterone and higher BMI. These results, although not establishing causality, suggest that maternal hormonal and metabolic status during embryo transfer may be associated with differences in implantation and early development depending on fetal sex. Neonatal sex ratio was analyzed solely as a biological outcome reflecting early developmental processes in ART.

Given the observational nature of this study, causal inference cannot be established. Although advanced statistical and causal inference methods were employed to mitigate confounding, the modest predictive performance and absence of dose–response gradients suggest that these findings should be viewed as exploratory and hypothesis-generating. The proposed biological pathways, ranging from progesterone-driven endometrial receptivity and embryo–endometrial synchrony to metabolic signaling and sex-specific implantation dynamics, remain theoretical and need validation through mechanistic and prospective studies.

Importantly, this study neither endorses nor encourages any form of sex selection or intentional modification of maternal physiology to determine fetal sex. It aims to enhance understanding of the natural interactions between maternal endocrine and metabolic factors and embryonic development in assisted reproduction.

Overall, our results emphasize the need for further research into how maternal physiology influences implantation success and secondary sex ratio outcomes in ART. While these findings do not warrant immediate changes in clinical practice, they may inform future studies to refine personalized reproductive care and elucidate the biological factors that influence sex-specific implantation.

## Author contributions

**Conceptualization:** Robert Czech, Przemysław Ciepiela.

**Data curation:** Robert Czech, Dariusz Wójcik, Tomasz Skweres, Wojciech Śliwiński, Dorota Zamkowska.

**Formal analysis:** Robert Czech, Przemysław Ciepiela.

**Investigation:** Robert Czech, Dariusz Wójcik, Tomasz Skweres, Wojciech Śliwiński, Dorota Zamkowska, Przemysław Ciepiela.

**Methodology:** Przemysław Ciepiela.

**Project administration:** Przemysław Ciepiela.

**Resources:** Robert Czech, Tomasz Skweres.

**Software:** Robert Czech.

**Supervision:** Dariusz Wójcik.

**Validation:** Przemysław Ciepiela.

**Writing – original draft:** Przemysław Ciepiela.

**Writing – review & editing:** Robert Czech, Przemysław Ciepiela.

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
