## [Decision Letter · Decision Letter 0]

23 Dec 2025

Dear Dr. Ciepiela,

Thank you for submitting your manuscript to PLOS ONE. After careful consideration, we feel that it has merit but does not fully meet PLOS ONE’s publication criteria as it currently stands. Therefore, we invite you to submit a revised version of the manuscript that addresses the points raised during the review process.

**ACADEMIC EDITOR: Please respond carefully for reviewers comments.**

We look forward to receiving your revised manuscript.

Kind regards,

Ayman A Swelum

Academic Editor

PLOS One

Journal Requirements:

3. We note you have included a table to which you do not refer in the text of your manuscript. Please ensure that you refer to Table 2 in your text; if accepted, production will need this reference to link the reader to the Table.

Reviewers' comments:

Reviewer's Responses to Questions

**Comments to the Author**

1. Is the manuscript technically sound, and do the data support the conclusions?

Reviewer #1: Partly

Reviewer #2: Yes

Reviewer #3: Partly

2. Has the statistical analysis been performed appropriately and rigorously?

Reviewer #1: Yes

Reviewer #2: Yes

Reviewer #3: Yes

3. Have the authors made all data underlying the findings in their manuscript fully available?

Reviewer #1: Yes

Reviewer #2: Yes

Reviewer #3: Yes

4. Is the manuscript presented in an intelligible fashion and written in standard English?

Reviewer #1: Yes

Reviewer #2: Yes

Reviewer #3: Yes

Reviewer #1: I appreciate the invitation to review this manuscript regarding the association between Progesterone, BMI, and the frequency of neonatal sex ratios. I would like to commend the authors for their research efforts and their submission to PLOS ONE.

The authors present an intriguing premise by correlating neonatal sex definition with variables measured after the embryos have already expressed their genetic sex. While the statistical associations found are noteworthy, the biological plausibility requires careful scrutiny. Given that the genetic sex of an embryo is immutable post-fertilization (specifically after the blastocyst stage), it is paramount that the methodology addresses the specific questions detailed below to clarify the proposed mechanism.

Introduction

The first two paragraphs of the Introduction should be condensed. They currently discuss factors influencing sex determination prior to blastocyst formation, which differs from the post-conceptional focus and hypothesis of the present study.

The explanatory content found in the final paragraph of the Introduction seems premature; I suggest moving these interpretations to the Discussion section to maintain a clear narrative flow.

Methodology

The methodology lacks data regarding BMI and progesterone measurements at the specific time of In Vitro Fertilization (IVF). Including these data points is essential to observe the variables present at the precise moment of genetic sex establishment.

The criteria for selecting the "first transferred embryo" are not clearly defined. Furthermore, the manuscript does not explain why subsequent embryo transfers were excluded from the report. According to the hypothesis presented, subsequent transfers should theoretically yield similar results; omitting them requires justification.

Male factor variables at the time of fertilization are not established in the current methodology. These must be included to rule out potential confounding factors that could influence sex ratios.

Results

Please explicitly state the absolute number (n) of male and female neonates within the "low" and "high" groups for the proposed Models 1 through 3. It is crucial to assess whether a small subset of patients with both low progesterone and low BMI might be disproportionately skewing the overall results of the models.

The implantation rates for both the low and high groups must be reported. This data is critical to support or refute the authors' hypotheses regarding differential survival or implantation based on sex.

Discussion

Please clarify whether there is any patient overlap between the current study and the self-referenced article cited in the discussion. Potential data duplication should be transparently addressed.

Concluding Observation

The authors posit a theory of lower implantation rates for female embryos in the presence of low BMI and low progesterone. Based on this premise, the most robust model to validate this hypothesis would be to determine and compare the implantation rates of embryos with previously known sex (e.g., via PGT-A) against the BMI and Progesterone variables. I recommend addressing why this approach was not taken or how the current model serves as a sufficient proxy.

Reviewer #2: It is a well-designed, informative, and relevant research paper that has sparked new discussions about the impact of maternal hormones and body composition on assisted reproductive technology (ART). Check the manuscript carefully as some corrections are made in the text.Groups need to be explained in detail. Table and graph numbers must be indicated while presenting the values in the result section. The diagnostic limitations should be made more explicit, especially the interpretation of the low AUC in the ROC and the caution in clinical application.

The ethical discussion section could be more comprehensively included, since sex determination is a sensitive social issue.

Suggestions for future research could be presented more clearly, such as a prospective cohort design and long-term outcome tracking, etc.

Reviewer #3: The manuscript serves interesting reading as it addresses clinically relevant question of the effect of maternal BMI and progesterone levels and neonatal sex ratio following the blastocyst transfer. In its current form however, the paper overstates causality of the relationship between those factors. It also does not display enough attention to the potential confounding role of oral dydrogesterone, which was part of the luteal support in studied patients. These issues significantly influence the interpretation of the findings. It is recommended to classify this manuscript as “major revision”

1. Throughout the whole manuscript the authors use language implying a causal relationship (e.g. “influence”, “affect”, “determine”) between serum progesterone/BMI and neonatal sex. Especially given the retrospective character of the study, such wording should be considered as overstatement and is not justified. Having used advanced statistical methodology (TMLE, IPW, E-values), still, it does not establish biological causality - particularly for an endpoint such as neonatal sex, reflecting a series of events (implantation, placentation, miscarriage, fetal development ). Additionally, no direct data are provided (nor are available) to support the described (sex-specific) effects of progesterone at implantation.

It is suggested to replace the causal language by associative terminology. The findings should be presented as hypothesis-generating rather than describing casual relationships.

2. Additionally, authors state that the endometrial preparation comprised of vaginal estradiol administered twice daily”, and for luteal support, vaginal micronised progesterone and oral dydrogesterone were used. Please specify the dosing of estradiol in the studied population and explain – rather untypical – vaginal route for its administration.

3. As per using oral dydrogesterone in the luteal support, it is worth noting that both the medication itself as well as its active metabolite are potent gestagens. Lack of cross-reactivity in progesterone assays does not imply lack of their biological effect.

Neumann et al. (Human Reproduction, 2022) demonstrated an inverse association between BMI and DHD concentrations, indicating that body composition and metabolism may modulate dydrogesterone exposure. Therefore, it cannot be excluded that part of the observed association between BMI, progesterone, and neonatal sex may be influenced by unmeasured variability in dydrogesterone/DHD exposure and it should be explicitly stated within the text.

It should be also included within the limitations section naming dydrogesterone as a potential confiounder / effect modifier of the observed associations between progesterone and SSR.

4. Suggested example changes in the text:

Line 46 – Conclusions

“Progesterone levels and BMI together influence neonatal sex ratios after frozen embryo transfer. These findings imply that endocrine and metabolic environments influence embryo–endometrium interactions in a sex-specific way and open new pathways for research into developmental programming in ART.”

Please consider changing to:

“Progesterone levels and BMI together were associated with neonatal sex ratios after frozen embryo transfer. These findings although not establishing causality, might imply that endocrine and metabolic environments influence embryo–endometrium interactions in a sex-specific way and open new pathways for research into developmental programming in ART”.

Line 274

“Although direct mechanistic data are limited, our results raise the possibility that circulating progesterone concentrations within physiologically relevant ranges for FET could selectively support the implantation of male or female embryos”

Please consider changing to :

“Although direct mechanistic data are limited, our results give rise to speculation that the possibility that circulating progesterone concentrations within physiologically relevant ranges for FET could selectively support the implantation of male or female embryos”

.

Reviewer #1: **Yes:** MD PhD Daniel Humberto Mendez LozanoMD PhD Daniel Humberto Mendez LozanoMD PhD Daniel Humberto Mendez LozanoMD PhD Daniel Humberto Mendez Lozano

Reviewer #2: No

Reviewer #3: No

---

## [Author Response · Author response to Decision Letter 1]

2 Feb 2026

Dear Academic Editor and Reviewers,

We sincerely thank the Academic Editor and all three Reviewers for their careful evaluation of our manuscript and their constructive, thoughtful, and detailed comments. We greatly appreciate the recognition of the study’s technical rigor, the appropriateness of the statistical analyses, and the relevance of the research question. In response to the reviewers’ feedback, we have undertaken a thorough revision of the manuscript to improve conceptual clarity, refine interpretive language, strengthen transparency, and explicitly address methodological and biological limitations, while preserving the integrity of the original analyses.

Below, we provide a point-by-point response to each reviewer’s comments. All revisions have been incorporated into the manuscript and are highlighted in the tracked-changes version. Line numbers refer to the Revised Manuscript with Track Changes.

Response to Reviewer #1

We thank Reviewer #1 for the thoughtful and detailed critique, particularly regarding biological interpretation, methodological clarity, and the study's framing.

Comment No. 1: The authors present an intriguing premise by correlating neonatal sex with variables measured after embryos have already expressed their genetic sex. While the statistical associations are noteworthy, the biological plausibility requires careful scrutiny. Given that the genetic sex of an embryo is immutable post-fertilization (specifically after the blastocyst stage), it is paramount that the methodology address the specific questions outlined below to clarify the proposed mechanism.

Response No. 1: We thank the Reviewer for this important conceptual comment and fully agree that genetic sex (XX/XY) is determined at fertilization and remains immutable thereafter. Importantly, our study does not propose that maternal serum progesterone levels or BMI determine embryonic sex. Rather, it examines whether variation in the maternal hormonal and metabolic environments during the peri-implantation period is associated with differences in the secondary sex ratio (SSR) observed among liveborn infants.

Our premise is that although genetic sex is fixed at fertilization, the sex ratio at birth may be modified by sex-specific differences in implantation success and/or early pregnancy survival that occur after fertilization. Progesterone plays a central role in endometrial receptivity, decidualization, immune modulation, and early placentation. Although these processes are not inherently sex-specific, accumulating evidence indicates that male and female embryos and placentas differ in early growth dynamics and stress responses, potentially leading to sex-differential vulnerability under suboptimal peri-implantation conditions. Accordingly, any observed association would reflect post-fertilization selection or survival rather than sex determination. To prevent misinterpretation, we have revised the manuscript to explicitly distinguish sex determination from variation in the secondary sex ratio and to consistently use non-causal, associative language throughout the text.

Action taken

Title revision: In response to concerns about conceptual clarity and potential overinterpretation, we revised the title to explicitly emphasize (i) the observational nature of the analysis (“associated with”), (ii) the population-level outcome (“neonatal sex ratios”), and (iii) the post-transfer timeframe.

Revised title: Maternal Serum Progesterone and BMI Are Associated with Neonatal Sex Ratios Following Single Frozen Embryo Transfer

Conceptual clarification in the Discussion (Lines 264–270):

“By isolating these variables within a standardized frozen embryo transfer setting, our aim was to generate hypotheses about how maternal metabolic and hormonal environments may relate to embryo–endometrium interactions and subsequent variation in the secondary sex ratio, rather than to imply the determination of genetic sex.”

Language refinement (Line 270):

Replaced causal phrasing (“affected outcomes”) with associative terminology (“were associated with neonatal sex outcomes”).

Explicit statement in the Conclusions (Lines 408–411):

Added clarification that the study does not support or enable sex selection and that the neonatal sex ratio was evaluated solely as a biological outcome reflecting early developmental and implantation-related processes.

Comment No. 2: Introduction. The first two paragraphs of the Introduction should be condensed. They currently discuss factors influencing sex determination before blastocyst formation, which differs from the post-conceptional focus and hypothesis of the present study.

The explanatory content in the final paragraph of the Introduction seems premature; I suggest moving these interpretations to the Discussion section to maintain a clear narrative flow.

Response No. 2: We agree with the Reviewer’s suggestion. The Introduction has been condensed to focus more directly on the study rationale, specifically the secondary sex ratio, implantation, and early pregnancy processes, while broader interpretive content has been moved to the Discussion. This restructuring improves narrative focus and ensures a clearer separation between background information and interpretation of findings.

Action taken: The first two paragraphs of the Introduction were shortened to reduce redundancy and limit interpretive statements. One interpretive sentence was moved from the final paragraph of the Introduction to the Discussion, where interpretation of the findings is more appropriate.

Relocated sentence (now Lines 264–268, Discussion):

“By isolating these variables within a standardized frozen embryo transfer setting, our aim was to generate hypotheses about how maternal metabolic and hormonal environments may relate to embryo–endometrium interactions and subsequent variation in the secondary sex ratio, rather than to imply determination of genetic sex.”

Comment No. 3: Methodology. The methodology lacks data on BMI and progesterone measurements at the specific time of In Vitro Fertilization (IVF). Including these data points is essential to capture the variables present at the precise moment of genetic sex establishment.

Response No. 3: We thank the Reviewer for this important comment and for highlighting the need to clarify the biological rationale for our hypothesis. We fully agree that genetic sex determination occurs at fertilization and remains immutable thereafter, and we emphasize that our study does not propose that maternal progesterone levels or BMI influence genetic sex determination. Instead, our hypothesis centers on a post-zygotic, implantation-stage selection mechanism. Specifically, we propose that maternal hormonal and metabolic conditions at the time of embryo transfer and implantation may be associated with sex-specific differences in implantation success and/or very early survival of embryos whose genetic sex has already been established. Under this framework, any observed differences in neonatal sex ratios reflect differential implantation or survival probabilities of male versus female embryos, rather than an alteration of genetic sex.

We acknowledge that this conceptual distinction may not have been sufficiently explicit in the original manuscript. We have therefore revised the Introduction and Methods to explicitly state that the biologically relevant exposure window in our analysis is the peri-implantation period, rather than fertilization or the establishment of genetic sex. We believe these revisions clarify the biological plausibility of our hypothesis while remaining fully consistent with established embryological principles.

Action taken:

Introduction (Lines 82–85):

Clarified that deviations in the secondary sex ratio are most plausibly influenced during post-fertilization development, implantation, or very early pregnancy rather than at fertilization.

Methods (Lines 107–112):

Explicitly explained why progesterone exposure was defined at the time of embryo transfer and why fertilization-stage measurements were not considered biologically informative for the study objective.

Discussion (Lines 289–292):

Clarified the biological relevance of progesterone concentrations measured around embryo transfer in relation to embryo–endometrium interactions and early survival processes.

Comment No. 4: The criteria for selecting the “first transferred embryo” are not clearly defined. Furthermore, the manuscript does not explain why subsequent embryo transfers were excluded from the report. According to the hypothesis presented, subsequent transfers should theoretically yield similar results; omitting them requires justification.

Response No. 4: We thank the Reviewer for this important methodological observation. Only the first frozen embryo transfer (FET) per patient was included to ensure statistical and biological independence of observations. Including multiple transfers from the same patient would introduce within-patient correlation and potential cumulative exposure and embryo selection biases, which could distort associations with the neonatal sex ratio.

Although subsequent transfers occur within the same individual, they are not biologically or analytically equivalent to the first transfer. Embryo hierarchy effects (e.g., preferential transfer of higher-quality embryos earlier), changes in endometrial conditions across cycles, and prior pregnancy outcomes may influence implantation dynamics and survival probabilities. Consequently, treating subsequent transfers as independent observations would violate core assumptions of standard regression models and could bias effect estimates.

We therefore restricted the analysis to the first FET per patient as a conservative design choice to maximize interpretability and methodological rigor. This approach is consistent with common practice in reproductive epidemiology when the outcome of interest is assessed at the patient level.

Action taken: Methods (Lines 113–115):

Added an explicit justification for including only the first frozen embryo transfer per patient:

“Only the first frozen embryo transfer per patient was included to avoid within-patient correlation, cumulative exposure bias, and embryo selection effects associated with multiple transfers.”

Comment No. 5: Male factor variables at the time of fertilization are not established in the current methodology. These must be included to rule out potential confounding factors that could influence sex ratios.

Response No. 5: We thank the Reviewer for raising this point. Male factor variables were considered in the analysis and included as covariates where available, specifically sperm concentration and motility. These parameters were included to account for potential male-related influences on reproductive outcomes. Importantly, existing evidence suggests that male factor infertility has a limited impact on the secondary sex ratio at live birth, particularly in assisted reproductive technology settings, especially in ICSI cycles, where sperm selection and fertilization procedures largely standardize fertilization conditions. As our study focuses on neonatal sex ratio rather than fertilization outcomes, residual male-related effects are expected to be minimal. Nonetheless, we acknowledge that unmeasured male factors cannot be entirely excluded. This limitation is now explicitly addressed in the manuscript to ensure transparency and appropriate interpretation of the findings.

Action taken: Methods (Lines 157–159):

Clarified the inclusion of male factor parameters (sperm concentration and motility) as covariates in the statistical models.

Limitations (Lines 382–385):

Acknowledged the possibility of residual confounding by unmeasured male factors while noting the limited expected impact on secondary sex ratio at live birth.

Comment No. 6: Results. Please explicitly state the absolute number (n) of male and female neonates within the “low” and “high” groups for the proposed Models 1 through 3. It is crucial to assess whether a small subset of patients with both low progesterone and low BMI might be disproportionately skewing the overall results of the models.

Response No. 6: We agree with the Reviewer that reporting absolute numbers is important for assessing the robustness of the findings. We have therefore added the absolute counts (n) of male and female neonates across the primary exposure strata for Models 1–3. These data show that the observed associations are not driven by a small subgroup of patients, including those with concomitant low progesterone and low BMI.

Action taken

Results section (Models 1–3): Added explicit reporting of the absolute numbers of male and female live births within the “low” and “high” exposure groups.

Table 2: Updated to include absolute counts (n) for male and female neonates for each exposure category.

Comment No. 7: The implantation rates for both the low and high groups must be reported. This data is critical to support or refute the authors’ hypotheses regarding differential survival or implantation based on sex.

Response No. 7: We agree that implantation and early survival are central to the biological interpretation of secondary sex ratio variation. However, because the present analysis was restricted to singleton live births, implantation failure and very early pregnancy loss were not directly observable in this dataset. Consequently, sex-specific implantation rates could not be calculated. As clarified in the manuscript, the neonatal sex ratio at birth was used as a proxy outcome reflecting the cumulative result of sex-specific implantation success and/or very early pregnancy survival. While this approach does not allow direct assessment of implantation rates, it is consistent with prior epidemiological studies examining secondary sex ratio using live-birth data. We now explicitly acknowledge this limitation and emphasize that the findings should be interpreted accordingly.

Action taken:

Introduction (Lines 81–84):

Clarified that implantation failure and very early pregnancy loss could not be directly assessed due to the study design, and justified the use of neonatal sex ratio as a proxy for sex-specific survival during implantation and early gestation.

Discussion:

Reiterated this limitation to ensure appropriate interpretation of the findings and to avoid overextension of mechanistic conclusions.

Comment No. 8: Please clarify whether there is any patient overlap between the current study and the self-referenced article cited in the discussion. Potential data duplication should be transparently addressed.

Response No. 8: We thank the Reviewer for raising this important point about transparency and potential data overlap. The database used in the present study is substantially larger than that used in our previously published work. Although some patients may appear in both datasets, the current analysis is not a duplication of prior results. Specifically, the present study includes a larger, updated cohort; a restriction to a more homogeneous, methodologically cleaner subgroup, namely patients undergoing their first frozen embryo transfer after a freeze-all strategy; and the incorporation of a new exposure variable, maternal serum progesterone concentration, which was not examined in the previous study. The analytical focus, study question, and primary exposures therefore differ meaningfully from the earlier report. Accordingly, the current work represents a distinct and independent analysis, designed to address a separate biological hypothesis, while benefiting from a refined study population that minimizes within-patient correlation and cumulative transfer bias.

Action taken: In the Discussion section, we added an explicit statement clarifying the relationship between the current study and the previously published work, emphasizing cohort expansion, refined inclusion criteria (first FET only), and the introduction of progesterone as a novel exposure variable.

Lines: 314 – 317: Although the dataset partially overlaps with a previously published cohort, the present study is an independent analysis of an expanded population limited to first frozen embryo transfer cycles and includes maternal serum progesterone conce

---

## [Decision Letter · Decision Letter 1]

19 Feb 2026

Dear Dr. Ciepiela,

Thank you for submitting your manuscript to PLOS ONE. After careful consideration, we feel that it has merit but does not fully meet PLOS ONE’s publication criteria as it currently stands. Therefore, we invite you to submit a revised version of the manuscript that addresses the points raised during the review process.

We look forward to receiving your revised manuscript.

Kind regards,

Ayman A Swelum

Academic Editor

PLOS One

**Journal Requirements:**

**Additional Editor Comments:**

Please respond carefully for all reviewer comments.

Reviewers' comments:

Reviewer's Responses to Questions

**Comments to the Author**

Reviewer #1: All comments have been addressed

Reviewer #2: All comments have been addressed

Reviewer #3: (No Response)

2. Is the manuscript technically sound, and do the data support the conclusions?

Reviewer #1: (No Response)

Reviewer #2: Yes

Reviewer #3: Yes

3. Has the statistical analysis been performed appropriately and rigorously?

Reviewer #1: (No Response)

Reviewer #2: Yes

Reviewer #3: Yes

4. Have the authors made all data underlying the findings in their manuscript fully available?

Reviewer #1: (No Response)

Reviewer #2: Yes

Reviewer #3: Yes

5. Is the manuscript presented in an intelligible fashion and written in standard English?

Reviewer #1: (No Response)

Reviewer #2: Yes

Reviewer #3: Yes

Reviewer #1: (No Response)

Reviewer #2: (No Response)

Reviewer #3: Thank you to the authors for submitting a revised version of the manuscript. The reviewers’ comments and suggested revisions have been incorporated into the text and have been appropriately addressed and explained in the rebuttal.

While the changes are generally satisfactory, some previously raised concerns remain regarding potential overstatements. Specifically, several formulations still read as overly suggestive of causality rather than reflecting the associative nature of the findings. Below, I provide suggested edits to the key passages in the Discussion. Once these are implemented, I recommend acceptance of the manuscript for publication.

Line No. 247

Was: Both factors were associated with neonatal sex outcomes, and their combination identified a maternal profile associated with a higher likelihood of male births.

Change to:Both factors were associated with neonatal sex outcomes, and their joint presence was associated with higher odds of male births.

Line No. 248

Was: Low PRG and low BMI each independently increased the likelihood of male offspring (Models 1 and 2), and their

combined presence further strengthened this association (Model 3).

Change to: Low PRG and low BMI were each independently associated with higher odds of male offspring (Models 1 and 2); when both were present, the odds were highest (Model 3).

Line No. 387

Was: A maternal profile with low

progesterone and low BMI was associated with a higher chance of male births, while higher progesterone and BMI more often correlated with female births.

Change to: Male births were more frequently observed among pregnancies with low progesterone and low BMI, while female births were more frequently observed among pregnancies with higher progesterone and higher BMI.

.

Reviewer #1: **Yes:** MD PhD Daniel Humberto Mendez LozanoMD PhD Daniel Humberto Mendez LozanoMD PhD Daniel Humberto Mendez LozanoMD PhD Daniel Humberto Mendez Lozano

Reviewer #2: **Yes:** Nasrin Sultana JuyenaNasrin Sultana JuyenaNasrin Sultana JuyenaNasrin Sultana Juyena

Reviewer #3: No

---

## [Author Response · Author response to Decision Letter 2]

22 Feb 2026

Dear Editor and Reviewers,

We would like to express our sincere gratitude for the time, expertise, and thoughtful consideration you devoted to reviewing our manuscript. Your insightful comments have been invaluable in improving the clarity, precision, and scientific rigor of our work. We truly appreciate the opportunity to revise the manuscript and are grateful for your constructive guidance, which has significantly strengthened the final version.

We believe that, as a result of your recommendations, the manuscript now presents a clearer and more methodologically rigorous account of our findings. In our view, the study provides novel, clinically relevant evidence in assisted reproductive technology, particularly regarding the association among maternal progesterone levels, BMI, and neonatal sex outcomes after frozen embryo transfer. We hope that the revised version better reflects both the scientific value and the associative nature of our findings. Below, we summarize the key revisions made in response to the remaining comments:

Response to Reviewer #2

As suggested, we revised the sentence in the Methods section to improve clarity and scientific precision. The original wording (Line 127):

“Senior embryologists documented morphology.”

has been replaced with:

“Senior embryologists prospectively recorded blastocyst morphological parameters in the laboratory database.”

Response to Reviewer #3

We sincerely thank Reviewer #3 for the careful evaluation and for highlighting the importance of avoiding causal overstatements. We have implemented the suggested edits verbatim to ensure that the language throughout the Discussion strictly reflects the associative nature of the findings. The following revisions were made exactly as recommended:

Line 247 revised to:

“Both factors were associated with neonatal sex outcomes, and their joint presence was associated with higher odds of male births.”

Line 248 revised to:

“Low PRG and low BMI were each independently associated with higher odds of male offspring (Models 1 and 2); when both were present, the odds were highest (Model 3).”

Line 387 revised to:

“Male births were more frequently observed among pregnancies with low progesterone and low BMI, while female births were more frequently observed among pregnancies with higher progesterone and higher BMI.”

In addition, we carefully reviewed the entire manuscript to ensure consistent use of associative rather than causal language. Finally, in accordance with the Editor’s instructions, we have included the following files with our resubmission:

A detailed point-by-point response to all comments raised by the Academic Editor and Reviewers, uploaded as a separate file labeled “Response to Reviewers.” A marked-up version of the manuscript highlighting all revisions, uploaded as “Revised Manuscript with Track Changes.” A clean, unmarked version of the revised manuscript, uploaded as “Manuscript.”

We are grateful for the opportunity to revise and resubmit our work and thank you again for your constructive feedback and consideration. We look forward to your decision.

Sincerely,

Przemysław Ciepiela, MD, PhD

---

## [Decision Letter · Decision Letter 2]

23 Mar 2026

Maternal Serum Progesterone and BMI Are Associated with Neonatal Sex Ratios Following Single Frozen Embryo Transfer

PONE-D-25-58987R2

Dear Dr. Ciepiela,

We’re pleased to inform you that your manuscript has been judged scientifically suitable for publication and will be formally accepted for publication once it meets all outstanding technical requirements.

Kind regards,

Ayman A Swelum

Academic Editor

PLOS One

Additional Editor Comments (optional):

Reviewers' comments:

Reviewer's Responses to Questions

**Comments to the Author**

Reviewer #3: All comments have been addressed

2. Is the manuscript technically sound, and do the data support the conclusions?

Reviewer #3: Yes

3. Has the statistical analysis been performed appropriately and rigorously?

Reviewer #3: Yes

4. Have the authors made all data underlying the findings in their manuscript fully available?

Reviewer #3: Yes

5. Is the manuscript presented in an intelligible fashion and written in standard English?

Reviewer #3: Yes

Reviewer #3: The reviewers' comments have been addressed in full. At this stage, as there are no further concerns, I recommend the manuscript for publication.

.

Reviewer #3: No

---

## [Editor Report · Acceptance letter]

PONE-D-25-58987R2

PLOS One

Dear Dr. Ciepiela,

I'm pleased to inform you that your manuscript has been deemed suitable for publication in PLOS One. Congratulations! Your manuscript is now being handed over to our production team.

Kind regards,

on behalf of

Professor Ayman A Swelum

Academic Editor

PLOS One